# DocAsRef: An Empirical Study on Repurposing Reference-based Summary Quality Metrics as Reference-free Metrics

**Forrest Sheng Bao** ⋈◇, **Ruixuan Tu**⋈♡, **Ge Luo**◇, **Yinfei Yang**♣,
**Hebi Li**◇, **Minghui Qiu**♠, **Youbiao He**◇, and **Cen Chen**¶

◇Department of Computer Science, Iowa State University, Ames, IA, USA
♡Dept. of Computer Sciences, University of Wisconsin–Madison, Madison, WI, USA
♣Sunnyvale, CA, USA
♠ByteDance, China
¶School of Data Science and Engineering, East China Normal University, Shanghai, China
⋈Equal contribution, `forrest.bao@gmail.com`, `ruixuan@cs.wisc.edu`

## Abstract

Automated summary quality assessment falls into two categories: reference-based and reference-free. Reference-based metrics, historically deemed more accurate due to the additional information provided by human-written references, are limited by their reliance on human input. In this paper, we hypothesize that the comparison methodologies used by some reference-based metrics to evaluate a system summary against its corresponding reference can be effectively adapted to assess it against its source document, thereby transforming these metrics into reference-free ones. Experimental results support this hypothesis. After being repurposed reference-freely, the zero-shot BERTScore using the pretrained DeBERTa-large-MNLI model of <0.5B parameters consistently outperforms its original reference-based version across various aspects on the SummEval and Newsroom datasets. It also excels in comparison to most existing reference-free metrics and closely competes with zero-shot summary evaluators based on GPT-3.5.

## 1 Introduction

Summarization is an important natural language generation (NLG) task. A problem that goes hand in hand with it is summary evaluation, which quantifies the quality of a summarizer or a system summary it generates. The traditional approach to automated[†] summary quality assessment is *reference-based*, such as ROUGE (Lin, 2004), BERTScore (Zhang* et al., 2020) and MoverScore (Zhao et al., 2019), which assesses a system summary against one or a plurality of human-written reference summaries.

Requiring highly educated human labor, reference summaries are very costly to obtain. Therefore, many *reference-free* metrics have emerged recently (Scialom et al., 2019; Vasilyev et al., 2020; Bao et al., 2022), which directly compute a score between a system summary and its source document. However, the performance of reference-free metrics has historically lagged behind that of reference-based metrics because a human-written reference summary serves as a fluent and comprehensive representation of the key facts in the input document and thus gives reference-based metrics an advantage.

Recently, large language models (LLMs) have shown promise in building reference-free summary quality metrics. Metrics based on LLMs like GPT-3.5/4 (Liu et al., 2023; Wang et al., 2023; Gao et al., 2023) have outperformed both reference-free and reference-based baselines. However, LLMs are computationally expensive, and the closed nature of GPT-3+ restricts their usage with legal and reproducibility[‡] limitations. A more viable solution that uses much more cost-effective language models is highly expected.

To build an accurate but efficient metric, we revisit the reference-based metrics and hypothesize that they can be repurposed into reference-free metrics by directly comparing a summary with its source document. After being repurposed, BERTScore outperforms not only its original reference-based version, but also most existing reference-free metrics across the SummEval, Newsroom, and TAC2010 datasets on both semantic and linguistic aspects. Notably, the repurposed BERTScore achieves superior or comparable per-

---

[†]The ground truth is still human evaluation.

[‡]`https://hackingsemantics.xyz/2023/closed-baselines/`

formance to GPT-3.5-based summarization evaluators. It is worth noting that these results are achieved using foundation models with significantly fewer parameters ($<0.5$B) compared to GPT-3.5's extensive 175 billion parameters.

We hope this paper can inspire more work into zero-shot summarization or NLG evaluation using cost-effective (e.g., $<1$B parameters) LMs. Our source code is at `https://github.com/SigmaWe/DocAsRef`. In summary, the key findings of this paper include:

1. The proposed reference-free repurposing does improve performances for Transformer-based metrics including BERTScore and BLEURT.

2. The repurposed BERTScore can significantly outperform all non-GPT-3.5 baselines using underlying LMs of the similar capacity.

3. With LMs hundreds of times smaller, the repurposed BERTScore can further match the performance of those based on GPT-3.5 in most of the cases.

## 2 Approach

### 2.1 Background: Ref-based and ref-free summary evaluation metrics

A system summary is generated from a source document by a summarizer, which is usually embodied by a neural network model today. A corresponding reference is generated from the same document by a human. Metrics for summary evaluation fall into two categories: the reference-based (short as *ref-based*) ones which are functions comparing a candidate summary and a human-written reference summary:
$$f(\text{system summary}, \text{reference}),$$
and reference-free (short as *ref-free*) ones which are functions that evaluate a candidate summary based solely on the input document:
$$f(\text{system summary}, \text{document}).$$
Ref-based metrics, such as ROUGE (Lin, 2004), BERTScore (Zhang* et al., 2020), BLEURT (Sellam et al., 2020), and MoverScore (Zhao et al., 2019), historically have an advantage over ref-free ones, such as Blanc (Vasilyev et al., 2020), SummQA (Scialom et al., 2019), SDC* (Liu et al., 2022), and SueNes (Bao et al., 2022), because the human-written reference summary serves as a fluent and comprehensive representation of the key facts in the input document. Recent GPT-based summary metrics (Gao et al., 2023; Wang et al., 2023; Liu et al., 2023) are all ref-free in nature.

### 2.2 Repurposing ref-based to ref-free

The idea of repurposing ref-based metrics for ref-free evaluation involves leveraging the mechanism employed by these metrics to compare two texts. Although ref-based metrics were originally designed to compare a system summary against a reference summary, we hypothesize that they can still be effective in directly comparing the system summary with the document.

To repurpose a ref-based metric $f$ into a ref-free one, we simply feed the document in lieu of the reference when using $f$. While the idea of using the document as the reference is not new, the specific approach proposed here, which is straightforward and direct, has not been previously explored. Embracing the principle that simplicity is beautiful in science, we decide to give it a try.

Remarkably, our simple strategy has yielded good results. Three representative ref-based metrics gain their performances after being repurposed (Table 1). One of them, BERTScore employing generically trained LMs such as RoBERTa-large has a performance very close to the performances of metrics based on GPT-3.5, which utilizes hundreds of times more parameters (Tables 2 & 3). This outcome highlights the effectiveness of repurposing ref-based metrics for ref-free evaluation.

### 2.3 Variants of BERTScore

The promising initial results encouraged us to explore modifications to the ref-based metrics for enhanced performances. ROUGE and BLEURT have limited room for tweaking because ROUGE-1 and ROUGE-2 have been the best among its variants in the past two decades and BLEURT is already fine-tuned explicitly for summary evaluation. Hence, we focus on refining BERTScore.

The first tweak we applied onto BERTScore is to try different small-scale, pretrained language models (LMs). We conducted experiments with three LMs: RoBERTa, DeBERTa, and BART, both their base versions (around 110M parameters) and large versions (around 400M parameters). Additionally, we explored the variants of these LMs that have been officially fine-tuned on the MNLI dataset. Our hypothesis is that an LM fine-tuned for the MNLI task may be better suited for computing text similarity than generic LMs.

The second tweak we explored is expanding BERTScore to the sentence level by calculating the similarity between sentences instead of tokens. Various similarity measures and sentence weight-

ing schemes were proposed (Appendix B). Unfortunately, they rarely perform better than the original token-level BERTScore.

## 3 Experiments

### 3.1 Settings

Because of their exceptional performances and impacts, four ref-based metrics are picked as **candidate metrics to be repurposed**: ROUGE (Lin, 2004), BERTScore (Zhang* et al., 2020), BLEURT (Sellam et al., 2020), and MoverScore (Zhao et al., 2019). ROUGE is the classic metric used in summarization. The rest three are widely used as baselines in the field in recent years.

Seven ref-free **baselines**[§] are included in our study. Four of them use underlying foundation LMs of fewer than 1B parameters: SummaQA (Scialom et al., 2019), BLANC (Vasilyev et al., 2020), SUPERT (Gao et al., 2020), and SueNes (Bao et al., 2022). The rest three (Liu et al., 2023; Gao et al., 2023; Wang et al., 2023) of them are based on GPT-3.5, which has 175B parameters.

Three multi-facet summarization evaluation datasets with human ratings are used as the **test datasets**: SummEval (Fabbri et al., 2021), Newsroom (Grusky et al., 2018) and TAC2010 (NIST, 2010). SummEval and Newsroom are for single-document summarization while TAC2010 is for multi-document summarization. SummEval covers four aspects: CONsistency, RELevance, COHerence, and FLUency. Newsroom covers four aspects: INFormativeness, RELevance, COHerence, and FLUency. TAC2010 reports three scores: Pyramid (Nenkova et al., 2007), linguistic, and overall scores. For TAC2010, only Set A of TAC2010 is used in this paper because Set B "update summarization" does not fit the problem formulation in § 2.1. Measuring how well a summary covers key pieces of information in the source document, RELevance or Pyramid score is generally considered the most important aspect of a summary. CONsistency a raising concern recently due to the hallucination issue. Details for the datasets and their aspects can be found from their respective papers.

**Underlying language models (LMs).** The LMs used in repurposed BERTScore variants are discussed in § 2.3. The default LM is RoBERTa-large. All ref-free baselines involving finetuning:

---

§We did not run the experiments on baselines but simply copied the numbers from their original papers to here. For the three GPT3.5-based baselines, we pick their best results from their papers.

BLANC, SummaQA, and SueNes, share the common initial checkpoint, BERT-base. MoverScore and BLUERT use RoBERTa-large and BLUERT-20 as the LMs.

BERTScore is a pairwise comparison metric. Depending on the axis along which max pooling is done, each BERTScore variant yields three scores: **P** (Precision), **R** (recall), and **F** (F1). The experiments are carried out on individual RTX 3090 24GB GPUs. For more details, see Appendix A.

### 3.2 Results

Following the trend in recent summary evaluation studies (Peyrard et al., 2017), we report the results at the summary level. Spearman's correlation coefficients between metrics' predictions and human-rated ground truth are the performance measure. For space sake, we present selected results here with extended results available in the appendices.

#### 3.2.1 Is repurposing useful? Before vs. after

The answer is yes! Despite that ref-based metrics historically perform better than ref-free metrics, Table 1 shows that the three modern metrics, MoverScore, BERTScore, and BLEURT, gain their performances after being repurposed, on nearly all aspects of all datasets. The lexicon-based ROUGE-1/2/L also improves its performance on some aspects or datasets after being repurposed.

After being repurposed (top of half of Table 1), BERTScore outperforms all other metrics across datasets, with only a couple of exceptions. It outperforms MoverScore and BLEURT significantly. While BERTScore underperforms ROUGE on SummEval before repurposing, it turns the tide after.

The ref-free metrics used in their original designated way perform extremely bad on the Newsroom dataset (bottom half of Table 1 and additional evidence in Appendix D). This is due to that in Newsroom, a reference summary can be as short as one sentence. Here, the reliance to reference summaries becomes a weakness of ref-based summary quality metrics. In this case, the original document may be better than the reference summary to compare with for judging the summary quality.

#### 3.2.2 Repurposed BERTScore vs. ref-free baselines

BERTScore is the most tweakable (§ 2.3) and best-performing (§ 3.2.1) metric. So we further study how it compares with the ref-free baselines. As mentioned in § 2.3, to study its robustness, different

Table 1: Performance before vs. after repurposing for four metrics. Summary-level. Spearman's. On the SummEval and Newsroom datasets. Best in each column in **bold** while 2nd best underlined.

| | | SummEval | | | | Newsroom | | | |
|---|---|---|---|---|---|---|---|---|---|
| | | CON | REL | COH | FLU | INF | REL | COH | FLU |
| After repurposing, used ref-freely | BERTScore P | **0.318** | 0.375 | **0.471** | **0.265** | 0.611 | 0.591 | 0.633 | 0.591 |
| | BERTScore R | 0.235 | 0.343 | 0.258 | 0.162 | **0.750** | **0.658** | 0.659 | 0.590 |
| | BERTScore F | 0.308 | **0.401** | 0.416 | 0.241 | 0.689 | 0.617 | **0.663** | **0.618** |
| | MoverScore | 0.180 | 0.245 | 0.138 | 0.093 | 0.695 | 0.615 | 0.589 | 0.537 |
| | ROUGE-1 R | 0.145 | 0.128 | 0.002 | 0.067 | 0.744 | 0.639 | 0.564 | 0.476 |
| | ROUGE-2 R | 0.262 | 0.155 | 0.049 | 0.163 | 0.746 | 0.648 | 0.591 | 0.511 |
| | ROUGE-L R | 0.289 | 0.187 | 0.106 | 0.183 | 0.746 | 0.641 | 0.591 | 0.515 |
| | BLEURT | 0.221 | 0.252 | 0.336 | 0.172 | 0.549 | 0.507 | 0.596 | 0.562 |
| Before repurposing, used in original ref-based way | BERTScore P | 0.008 | 0.208 | 0.275 | 0.083 | -0.034 | 0.012 | 0.044 | 0.045 |
| | BERTScore R | 0.158 | 0.355 | 0.284 | 0.148 | 0.315 | 0.294 | 0.311 | 0.320 |
| | BERTScore F | 0.088 | 0.301 | 0.321 | 0.139 | 0.149 | 0.171 | 0.185 | 0.187 |
| | MoverScore | 0.129 | 0.238 | 0.088 | 0.096 | 0.136 | 0.153 | 0.112 | 0.077 |
| | ROUGE-1 R | 0.148 | 0.250 | 0.117 | 0.109 | 0.105 | 0.128 | 0.071 | 0.073 |
| | ROUGE-2 R | 0.166 | 0.194 | 0.109 | 0.102 | 0.069 | 0.087 | 0.016 | 0.037 |
| | ROUGE-L R | 0.123 | 0.205 | 0.146 | 0.099 | 0.035 | 0.063 | 0.016 | 0.025 |
| | BLEURT | 0.048 | 0.215 | 0.174 | 0.087 | 0.154 | 0.140 | 0.071 | 0.075 |

underlying LMs are used with BERTScore. Due to space limit, here we only report the results using RoBERTa-large and DeBERTa-large which give the best performance.

Table 2: Summary-level Spearman's correlation coefficients on dataset SummEval. Aspect names abbreviated.

| | CON | REL | COH | FLU |
|---|---|---|---|---|
| *BERTScore, repurposed, using respective LMs below* | | | | |
| RoBERTa-large P | 0.318 | 0.375 | **0.471** | 0.265 |
| RoBERTa-large F | 0.308 | 0.401 | 0.416 | 0.241 |
| RoBERTa-large-MNLI P | 0.387 | 0.358 | 0.438 | 0.287 |
| RoBERTa-large-MNLI F | 0.357 | 0.382 | 0.373 | 0.241 |
| DeBERTa-large P | 0.338 | 0.341 | 0.418 | 0.280 |
| DeBERTa-large F | 0.289 | 0.357 | 0.315 | 0.211 |
| DeBERTa-large-MNLI P | 0.399 | 0.293 | 0.351 | 0.303 |
| DeBERTa-large-MNLI F | 0.344 | 0.333 | 0.291 | 0.239 |
| Best of Repurposed BERTScore | 0.399 | 0.401 | 0.471 | 0.303 |
| *Baselines, reference-free* | | | | |
| Blanc | 0.244 | 0.197 | 0.089 | 0.132 |
| SummaQA-F1 | 0.197 | 0.165 | 0.123 | 0.140 |
| SUPERT | 0.330 | 0.216 | 0.120 | 0.230 |
| SueNes | 0.190 | 0.177 | 0.167 | 0.228 |
| ChatGPT (Wang et al., 2023) | **0.432** | **0.428** | 0.470 | 0.353 |
| G-Eval (GPT-3.5) (Liu et al., 2023) | 0.386 | 0.385 | 0.440 | **0.424** |
| Best of Baselines | 0.432 | 0.428 | 0.470 | 0.424 |

The results on SummEval are given in Table 2. Repurposed BERTScore outperforms all non-GPT baselines by a significant margin. Additionally, it performs comparably to GPT3.5-based baselines on the RELevance and COHerence aspects. It is superior than one of the two GPT-3.5-based approaches on the CONsistency aspect. It should be noted that SummEval is challenging due to its coverage of 23 modern summarizers, many of which exhibit highly similar behavior.

Table 3 reports the results on the Newsroom dataset. The Newsroom dataset poses a significant challenge for new metrics since the baselines already perform very well on this dataset, likely because it evaluates only seven systems with dis-

Table 3: Summary-level Spearman's correlation coefficients on dataset Newsroom. Aspect names abbreviated.

| | INF | REL | COH | FLU |
|---|---|---|---|---|
| *BERTScore, repurposed, using respective LMs below* | | | | |
| RoBERTa-large R | 0.750 | **0.658** | 0.659 | 0.590 |
| RoBERTa-large F | 0.689 | 0.617 | 0.663 | 0.618 |
| RoBERTa-large-MNLI R | 0.737 | 0.621 | 0.632 | 0.550 |
| RoBERTa-large-MNLI F | 0.680 | 0.582 | 0.641 | 0.563 |
| DeBERTa-large R | 0.747 | 0.646 | **0.669** | 0.604 |
| DeBERTa-large F | 0.720 | 0.625 | 0.676 | 0.613 |
| DeBERTa-large-MNLI R | 0.748 | 0.629 | 0.668 | 0.583 |
| DeBERTa-large-MNLI F | 0.739 | 0.635 | 0.674 | 0.595 |
| Best of Repurposed BERTScore | 0.750 | 0.658 | 0.669 | 0.618 |
| *Baselines, reference-free* | | | | |
| Blanc | 0.688 | 0.608 | 0.586 | 0.531 |
| SummaQA-F1 | 0.569 | 0.516 | 0.490 | 0.466 |
| SUPERT | 0.693 | 0.605 | 0.617 | 0.539 |
| SueNes | **0.753** | 0.647 | **0.669** | **0.674** |
| ChatGPT (Gao et al., 2023) | 0.521 | 0.524 | 0.484 | 0.480 |
| ChatGPT (Wang et al., 2023) | 0.578 | 0.461 | 0.469 | 0.507 |
| Best of Baselines | 0.753 | 0.647 | 0.669 | 0.674 |

tinct performances. Despite the challenges, repurposed BERTScore outperforms all baselines except SueNes, which is finetued using data explicitly augmented for the summary evaluation task, on all aspects.

Because the non-GPT baselines, BLANC, SummaQA, and SueNes, use BERT-base as the underlying LM, for a fair comparison, we include BERTScore's results using RoBERTa/DeBERTa/BART-base in Appendix D. Even when they use LMs of the same size, BERTScore still outperforms them.

Table 4 shows the results on the TAC2010 dataset where BERTScore outperforms baselines on all aspects except linguistics. As a multi-document summarization dataset, TAC2010 provides 10 source documents $d_1, \cdots, d_{10}$ for generating a system summary $s$. We use the formula $\sum_{i \in [1..10]} f(d_i, s)$ to approximate the score of a

Table 4: Summary-level Spearman's correlation coefficients on dataset TAC2010 (multi-document summarization). Aspect names in table header.

| | Pyramid | Linguistic | Overall |
|---|---|---|---|
| *BERTScore, repurposed, using respective LMs below* | | | |
| DeBERTa-large-MNLI P | 0.496 | 0.401 | 0.455 |
| DeBERTa-large-MNLI R | 0.526 | 0.405 | 0.492 |
| DeBERTa-large-MNLI F | **0.539** | 0.422 | **0.500** |
| BART-large-MNLI P | 0.471 | 0.272 | 0.415 |
| BART-large-MNLI R | 0.422 | 0.202 | 0.380 |
| BART-large-MNLI F | 0.481 | 0.245 | 0.426 |
| RoBERTa-large-MNLI P | 0.469 | 0.306 | 0.418 |
| RoBERTa-large-MNLI R | 0.481 | 0.340 | 0.450 |
| RoBERTa-large-MNLI F | 0.509 | 0.356 | 0.464 |
| | | | |
| *Baselines, reference-free* | | | |
| Blanc | 0.427 | 0.294 | 0.397 |
| SummaQA-F1 | 0.301 | 0.243 | 0.286 |
| SUPERT | 0.479 | 0.324 | 0.427 |
| SueNes | 0.492 | **0.460** | 0.470 |

summary $s$ given a single-document summarization metric $f$.

## 3.3 What makes BERTScore powerful

While the result of this paper may sound surprising because the method is very simple, it is totally explainable. Comparing a summary with a document is theoretically more challenging than comparing it with a reference, because information is more sparse in a document than in a reference. This might be the reason that strong NLG evaluation metrics are historically reference-based. However, BERTScore exhibits exceptional performance after being repurposed from ref-based to ref-free. We attribute this to both the contextual embedding of the underlying LMs and the maxpooling step of BERTScore.

The Transformers have the ability to identify important information in a context: by showing strong attentions to the important tokens as learned in pretraining. In other words, encoder-only Transformers used in BERTScore can identify important tokens and function as implicit summarizers. Extraneous information in a summary causes the summary's context to diverge from that of the original document, resulting in a reduction of semantic similarity, even when comparing the same token in the summary to its 'counterpart in the document. The maxpooling step of BERTScore further focuses on the alignment of the most semantically proximate token pairs between the document and the summary. Because the document and the summary are independently embedded in BERTScore, only when important information in the document and the summary align, the BERTScore can be high. On a related note, BERTScore alone is found very

effectively in measuring factual inconsistency in summaries (Laban et al., 2022).

Table 5: The performance of BERTScore-P with and without IDF. Summary-level Spearman's correlation coefficients in comparison. Model size: base. Yellow cells are when using IDF is worse than without IDF and green cells are for the opposite.

| | | SummEval | | | | Newsroom | | | |
|---|---|---|---|---|---|---|---|---|---|
| | | CON | REL | COH | FLU | INF | REL | COH | FLU |
| IDF on | RoBERTa | 0.295 | 0.284 | 0.381 | 0.228 | 0.627 | 0.579 | 0.589 | 0.536 |
| | BART | 0.279 | 0.283 | 0.359 | 0.208 | 0.673 | 0.631 | 0.664 | 0.620 |
| | DeBERTa | 0.262 | 0.252 | 0.316 | 0.206 | 0.614 | 0.556 | 0.613 | 0.544 |
| IDF off | RoBERTa | 0.307 | 0.315 | 0.408 | 0.240 | 0.597 | 0.551 | 0.579 | 0.531 |
| | BART | 0.291 | 0.322 | 0.390 | 0.233 | 0.675 | 0.650 | 0.661 | 0.610 |
| | DeBERTa | 0.281 | 0.276 | 0.345 | 0.221 | 0.628 | 0.587 | 0.631 | 0.586 |

The IDF part of BERTScore may not play an important role because the attention mechanism already factors in what IDF does. A stopword or a boilerplate word has a weak attention to other tokens. In BERTScore's original paper (Zhang* et al., 2020), IDF makes very marginal impact on all except one datasets/tasks. Table 5 shows our ablation study on the impact of IDF. IDF makes a very small impact and in many cases, it even decreases the performance.

The repurposed BERTScore shows relatively robust performance with respect to the choice of the underlying LMs. For example, on Newroom, BERTScore's worst performing variant in every aspect still outperforms the ChatGPT-based. The only aspect on which BERTScore is not stable is the COHerence aspect of SummEval.

## 4 Conclusion

In this paper, we explore repurposing summary evaluation metrics that were originally designed or trained for reference-based use as reference-free metrics. The motivation was to reuse their power in comparing texts. Comprehensive experiments on multiple datasets show that four representative metrics generally perform better after the repurposing. The best among them, BERTScore, is further studied with different configurations. The repurposed BERTScore using 0.5B-parameter LMs can outperform all non-GPT baselines significantly and even most of the times those based on GPT3.5.

## Acknowledgments

This research is partially supported by National Science Foundation (NSF) grant CNS-1817089. The authors would also like to thank reviewers who have given precious feedback on improving this

work. Forrest Bao also wants to dedicate this paper to the people of Ukraine who have been courageously fighting for freedom since February 24, 2022.

## Limitations

The test sets are all from the news domain which is the only domain that human evaluation to system summaries has been done. This is a limit beyond our control.

Unfortunately, our attempt (Appendix B) to expand BERTScore from token-level to sentence-level fails. Moreover, unlike token-level BERTScore, which remains stable across different LM choices, sentence-level BERTScore is highly sensitive to the selection of LMs. Extended results can be found in the appendices.

BERTScore can have a variant, which is at chuck-level. This idea was proposed in REUSE for machine translation (Mukherjee and Shrivastava, 2022). Since we have tried token-level and sentence-level BERTScore, trying chuck-level BERTScore in summarization can be part of the future work.

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

## A More Experimental Information

More details on **underline language models**, BLANC and SueNes, the two training-based reference-free baselines use BERT-base while the training-based reference-based BLEURT uses BLEURT-20[¶], a 32-layer Transformer model. SUPERT uses BERT-Large-based SentenceTransformer while SummaQA uses a BERT-Large-based QA model. Please understand the tremendous amount of effort and time needed to re-train or re-benchmark all metrics using the same underlying pre-trained language model.

**Experimental time:** The experiment can be done really quickly. For SummEval, 5 mins for

---

[¶]Per the BLEURT authors, BLEURT-20 is the strongest, released BLEURT model https://github.com/google-research/bleurt/blob/master/checkpoints.md#the-recommended-checkpoint-bleurt-20

each token-level metric, 30 mins for each sentence-level BERTScore-variant. For Newsroom, the numbers are 8 minutes and 45 minutes, respectively.

**Software packages:** We used HuggingFace's `evaluate` library for the metrics and `sentence-transformer` library for cosine similarity computation. The `evaluate` library automatically downloads and plugs models into the metrics. We also used `numpy` and `scipy` for general computation. For MoverScore, we used its official implementation from its Github Repo https://github.com/AIPHES/emnlp19-moverscore. The v1 code has deprecated dependencies. So we used the v2 version.

## B Expanding BERTScore to the sentence level

We experimented with two approaches to measure sentence similarity: cosine/dot-product similarity and using text reasoning probabilities. Let us elaborate on the latter. Models trained for natural language inference (NLI) tasks typically output three probabilities representing the relationships between two input sentences: "entailing" (E), "neutral" (N), and "contradictory" (C). In other words, given a pair of sentences $x$ and $y$, we obtain: $[E, N, C] = \text{NLI}(x, y)$. We experimented with three options: $1 - N$, $E - C$, and $E$, and selected $E - C$ due to its intuitive appeal and empirical evidence of its effectiveness.

The original BERTScore uses IDF to weight tokens. To weight sentences, we employ a PageRank-style approach below. First, we decide the importance of document sentences $[x_1, x_2, \ldots]$. A sentence is considered important if it can relate to many other sentences. Hence, the importance of a document sentence $x_i$ can be estimated as $w_i = g(\text{sim}(x_i, x_1), \text{sim}(x_i, x_2), \ldots)$ where $\text{sim}(\cdot)$ is a sentence-level similarity measure and $g$ can be sum or entropy. In the simplest case, we have $w_i = \sum_{i \neq j, j \in \mathbb{N}^+} \text{sim}(x_i, x_j)$. Second, we let the document sentences "vote" on the importance of summary sentences. The importance of a summary sentence is determined by the sum of its similarities to all document sentences, weighted by the importance (voting power) of document sentences. Thus, the importance of the $j$-th sentence $y_j$ in the summary is $v_j = \sum_{i, j \in \mathbb{N}^+} w_i \text{sim}(x_i, y_j)$.

Unfortunately, as you can see in § D, the sentence-level tweaks do not yield better results except on the consistency aspect. Since there are too

many sentence-level BERTScore variants, they are referred to in this A-B-C nomenclature, where A is the similarity measure which is `Cosine` if cosine similarity and `MNLI` if using entailment confidence from an MNLI-finetuned model, B is the underlying LM, and C, optional, is the sentence weighting method $g$.

## C   Our idea in code

We hope this code can help explain what we mean by "repurposing" and also how to directly use the conclusion of this paper.

```python
import evaluate   # HuggingFace's
import functools  # Python's standard

bertscore = evaluate.load("bertscore")
bertscore_deberta_large_mnli = functools.partial(
    bertscore.compute,
    lang="en",
    use_fast_tokenizer=True,
    model_type="microsoft/deberta-large-mnli"
)
scores = bertscore_deberta_large_mnli(
    predictions = ["this is a summary"],
  # references = ["this is a reference"] # old way
    references = ["this is the DOC"] # DocAsRef
)[0]['recall']
```

At line 13, conventional approaches plug in human-written references. But in our proposed idea (line 14), just plug in the source documents, and you will get the best reference-free summary quality assessor.

It's easy-to-implement, zero-shot, and reference-free.

## D   More comprehensive results

Please refer to Table 6 and Table 7.

## E   Leadword heuristic

It is common that important information is clustered at the beginning of a document. Hence, Leadword is a simple but effective method to extract important information. SUPERT build pseudo-references by extracting salient sentences and found that Leadword is better than any other simple extractive approach. So we also experiment with limiting the BERTScore-style pairwise comparison to top-$k$ sentences in the input document. We use top-$k$ slightly different from its common use in text generation. Here $k$ means a ratio rather than an absolute number because the length of the input document varies a lot.

**Is Leadword heuristic useful?** In this study, no repurposed metrics benefit from the Leadword heuristic, unlike the result reported in SUPERT (Gao et al., 2020). Nearly every metric loses

performance after using the Leadword heuristic. The shorter the lead is, the more performance drop. Investigating the reason is part of our future work.

Table 6: Extended Spearman results.

| | SummEval | | | | Newsroom | | | |
|---|---|---|---|---|---|---|---|---|
| | CONsistency | RELevance | COHerence | FLUency | INFormativeness | RELevance | COHerence | FLUency |
| **BERTScore, token-level, repurposed, using respective LMs below** | | | | | | | | |
| RoBERTa-base P | 0.307 | 0.315 | 0.408 | 0.240 | 0.597 | 0.551 | 0.579 | 0.531 |
| RoBERTa-base R | 0.179 | 0.282 | 0.196 | 0.108 | 0.739 | 0.632 | 0.616 | 0.540 |
| RoBERTa-base F | 0.278 | 0.336 | 0.339 | 0.200 | 0.692 | 0.606 | 0.626 | 0.556 |
| DeBERTa-base P | 0.281 | 0.276 | 0.345 | 0.221 | 0.628 | 0.587 | 0.631 | 0.586 |
| DeBERTa-base R | 0.204 | 0.309 | 0.207 | 0.132 | 0.736 | 0.635 | 0.637 | 0.575 |
| DeBERTa-base F | 0.263 | 0.343 | 0.296 | 0.191 | 0.720 | 0.626 | 0.662 | 0.588 |
| BART-base P | 0.291 | 0.322 | 0.390 | 0.233 | 0.675 | 0.650 | 0.661 | 0.610 |
| BART-base R | 0.147 | 0.268 | 0.176 | 0.057 | 0.752 | 0.650 | 0.621 | 0.561 |
| BART-base F | 0.218 | 0.321 | 0.260 | 0.128 | 0.765 | 0.664 | 0.679 | 0.617 |
| RoBERTa-large P | 0.318 | 0.375 | 0.471 | 0.265 | 0.611 | 0.591 | 0.633 | 0.591 |
| RoBERTa-large R | 0.235 | 0.343 | 0.258 | 0.162 | 0.750 | 0.658 | 0.659 | 0.590 |
| RoBERTa-large F | 0.308 | 0.401 | 0.416 | 0.241 | 0.689 | 0.617 | 0.663 | 0.618 |
| RoBERTa-large-MNLI P | 0.387 | 0.358 | 0.438 | 0.287 | 0.617 | 0.554 | 0.609 | 0.548 |
| RoBERTa-large-MNLI R | 0.264 | 0.327 | 0.241 | 0.155 | 0.737 | 0.621 | 0.632 | 0.550 |
| RoBERTa-large-MNLI F | 0.357 | 0.382 | 0.373 | 0.241 | 0.680 | 0.582 | 0.641 | 0.563 |
| DeBERTa-large P | 0.338 | 0.341 | 0.418 | 0.280 | 0.650 | 0.596 | 0.651 | 0.616 |
| DeBERTa-large R | 0.222 | 0.310 | 0.225 | 0.138 | 0.747 | 0.646 | 0.669 | 0.604 |
| DeBERTa-large F | 0.289 | 0.357 | 0.315 | 0.211 | 0.720 | 0.625 | 0.676 | 0.613 |
| DeBERTa-large-MNLI P | 0.399 | 0.293 | 0.351 | 0.303 | 0.642 | 0.594 | 0.639 | 0.605 |
| DeBERTa-large-MNLI R | 0.271 | 0.305 | 0.220 | 0.183 | 0.748 | 0.629 | 0.668 | 0.583 |
| DeBERTa-large-MNLI F | 0.344 | 0.333 | 0.291 | 0.239 | 0.739 | 0.635 | 0.674 | 0.595 |
| BART-large P | 0.299 | 0.350 | 0.397 | 0.226 | 0.701 | 0.621 | 0.699 | 0.656 |
| BART-large R | 0.186 | 0.294 | 0.199 | 0.098 | 0.758 | 0.657 | 0.633 | 0.584 |
| BART-large F | 0.245 | 0.345 | 0.279 | 0.163 | 0.768 | 0.651 | 0.682 | 0.615 |
| BART-large-MNLI P | 0.336 | 0.355 | 0.421 | 0.267 | 0.676 | 0.613 | 0.672 | 0.644 |
| BART-large-MNLI R | 0.205 | 0.300 | 0.193 | 0.116 | 0.764 | 0.654 | 0.619 | 0.560 |
| BART-large-MNLI F | 0.282 | 0.360 | 0.289 | 0.186 | 0.773 | 0.655 | 0.670 | 0.621 |
| Best of Repurposed BERTScore | 0.399 | 0.401 | 0.471 | 0.303 | 0.773 | 0.664 | 0.699 | 0.656 |
| **BERTScore, sentence-level, repurposed, using respective LMs below** | | | | | | | | |
| Cos. MPNet-base P | 0.378 | 0.169 | 0.210 | 0.315 | 0.565 | 0.578 | 0.613 | 0.612 |
| Cos. MPNet-base R | 0.182 | 0.207 | 0.093 | 0.097 | 0.658 | 0.557 | 0.554 | 0.503 |
| Cos. MPNet-base F | 0.322 | 0.218 | 0.156 | 0.220 | 0.687 | 0.599 | 0.629 | 0.594 |
| Cos. MPNet-base Sum-wt P | 0.386 | 0.170 | 0.216 | 0.315 | 0.592 | 0.587 | 0.636 | 0.613 |
| Cos. MPNet-base Sum-wt R | 0.287 | 0.232 | 0.130 | 0.204 | 0.679 | 0.598 | 0.613 | 0.596 |
| Cos. MPNet-base Sum-wt F | 0.357 | 0.218 | 0.182 | 0.274 | 0.695 | 0.611 | 0.674 | 0.653 |
| MNLI DeBERTa-large-MNLI P | 0.395 | 0.152 | 0.154 | 0.319 | 0.318 | 0.353 | 0.398 | 0.428 |
| MNLI DeBERTa-large-MNLI R | 0.141 | 0.130 | -0.047 | 0.092 | 0.431 | 0.356 | 0.428 | 0.396 |
| MNLI DeBERTa-large-MNLI F | 0.179 | 0.140 | -0.026 | 0.117 | 0.341 | 0.270 | 0.312 | 0.222 |
| MNLI DeBERTa-large-MNLI Entropy-wt P | 0.409 | 0.160 | 0.171 | 0.323 | 0.264 | 0.319 | 0.354 | 0.387 |
| MNLI DeBERTa-large-MNLI Entropy-wt R | 0.002 | 0.048 | -0.002 | -0.026 | 0.174 | 0.127 | 0.271 | 0.248 |
| MNLI DeBERTa-large-MNLI Entropy-wt F | -0.120 | 0.001 | -0.015 | -0.115 | -0.031 | -0.076 | 0.019 | -0.046 |
| **Repurposed other metrics** | | | | | | | | |
| ROUGE-1 R | 0.145 | 0.128 | 0.002 | 0.067 | 0.744 | 0.639 | 0.564 | 0.476 |
| ROUGE-2 R | 0.262 | 0.155 | 0.049 | 0.163 | 0.746 | 0.648 | 0.591 | 0.511 |
| ROUGE-L R | 0.289 | 0.187 | 0.106 | 0.183 | 0.746 | 0.641 | 0.591 | 0.515 |
| BLEURT | 0.221 | 0.252 | 0.336 | 0.172 | 0.549 | 0.507 | 0.596 | 0.562 |
| MoverScore | 0.180 | 0.245 | 0.138 | 0.093 | 0.695 | 0.615 | 0.589 | 0.537 |
| **Baselines, other reference-free metrics** | | | | | | | | |
| Blanc | 0.244 | 0.197 | 0.089 | 0.132 | 0.688 | 0.608 | 0.586 | 0.531 |
| SummaQA-F1 | 0.197 | 0.165 | 0.123 | 0.140 | 0.569 | 0.516 | 0.490 | 0.466 |
| SUPERT | 0.330 | 0.216 | 0.120 | 0.230 | 0.693 | 0.605 | 0.617 | 0.539 |
| SueNes | 0.190 | 0.177 | 0.167 | 0.228 | 0.753 | 0.647 | 0.669 | 0.674 |
| ChatGPT (Gao et al., 2023) | 0.435 | 0.433 | 0.561 | 0.419 | 0.521 | 0.524 | 0.484 | 0.480 |
| ChatGPT(Wang et al., 2023) | 0.432 | 0.439 | 0.451 | 0.380 | 0.578 | 0.461 | 0.469 | 0.507 |
| G-Eval (GPT-3.5) (Liu et al., 2023) | 0.386 | 0.385 | 0.440 | 0.424 | NA | NA | NA | NA |
| SDC* (Liu et al., 2022) | -0.080 | -0.068 | 0.062 | 0.002 | -0.708 | -0.627 | -0.536 | -0.453 |
| **Reference-based approaches used in their original designated way** | | | | | | | | |
| ROUGE-1 R | 0.154 | 0.309 | 0.164 | 0.117 | 0.323 | 0.278 | 0.231 | 0.215 |
| ROUGE-2 R | 0.178 | 0.272 | 0.182 | 0.142 | 0.153 | 0.134 | 0.086 | 0.102 |
| ROUGE-L R | 0.111 | 0.296 | 0.119 | 0.109 | 0.301 | 0.263 | 0.206 | 0.201 |
| MoverScore | 0.195 | 0.299 | 0.175 | 0.185 | 0.219 | 0.216 | 0.174 | 0.143 |
| BERTScore RoBERTa-base P | -0.020 | 0.171 | 0.223 | 0.045 | -0.071 | -0.007 | -0.018 | -0.026 |
| BERTScore RoBERTa-base P | 0.134 | 0.352 | 0.252 | 0.124 | 0.320 | 0.295 | 0.285 | 0.268 |
| BERTScore RoBERTa-base F | 0.059 | 0.282 | 0.270 | 0.094 | 0.125 | 0.162 | 0.139 | 0.137 |
| BERTScore RoBERTa-large P | 0.008 | 0.208 | 0.275 | 0.083 | -0.034 | 0.012 | 0.044 | 0.045 |
| BERTScore RoBERTa-large R | 0.158 | 0.355 | 0.284 | 0.148 | 0.315 | 0.294 | 0.311 | 0.320 |
| BERTScore RoBERTa-large F | 0.088 | 0.301 | 0.321 | 0.139 | 0.149 | 0.171 | 0.185 | 0.187 |
| BLEURT | 0.163 | 0.272 | 0.163 | 0.191 | 0.316 | 0.282 | 0.271 | 0.239 |
| METEOR (Banerjee and Lavie, 2005) | 0.170 | 0.253 | 0.120 | 0.124 | 0.242 | 0.223 | 0.163 | 0.173 |

Table 7: Extended Pearson results.

| | SummEval | | | | Newsroom | | | |
|---|---|---|---|---|---|---|---|---|
| | CONsistency | RELevance | COHerence | FLUency | INFormativeness | RELevance | COHerence | FLUency |
| **BERTScore, token-level, repurposed, using respective LMs below** | | | | | | | | |
| RoBERTa-base P | 0.312 | 0.321 | 0.420 | 0.269 | 0.664 | 0.661 | 0.615 | 0.554 |
| RoBERTa-base R | 0.172 | 0.284 | 0.194 | 0.095 | 0.805 | 0.753 | 0.688 | 0.630 |
| RoBERTa-base F | 0.282 | 0.345 | 0.357 | 0.217 | 0.750 | 0.725 | 0.669 | 0.606 |
| DeBERTa-base P | 0.317 | 0.294 | 0.349 | 0.271 | 0.711 | 0.703 | 0.657 | 0.586 |
| DeBERTa-base R | 0.206 | 0.317 | 0.185 | 0.123 | 0.809 | 0.766 | 0.703 | 0.641 |
| DeBERTa-base F | 0.285 | 0.347 | 0.288 | 0.208 | 0.786 | 0.756 | 0.701 | 0.631 |
| BART-base P | 0.306 | 0.331 | 0.408 | 0.243 | 0.720 | 0.720 | 0.678 | 0.614 |
| BART-base R | 0.134 | 0.267 | 0.154 | 0.033 | 0.818 | 0.775 | 0.699 | 0.637 |
| BART-base F | 0.219 | 0.322 | 0.262 | 0.122 | 0.815 | 0.783 | 0.719 | 0.651 |
| RoBERTa-large P | 0.343 | 0.377 | 0.483 | 0.316 | 0.684 | 0.682 | 0.646 | 0.590 |
| RoBERTa-large R | 0.241 | 0.348 | 0.248 | 0.161 | 0.803 | 0.746 | 0.714 | 0.660 |
| RoBERTa-large F | 0.337 | 0.411 | 0.425 | 0.277 | 0.749 | 0.725 | 0.688 | 0.630 |
| RoBERTa-large-MNLI P | 0.440 | 0.386 | 0.467 | 0.369 | 0.686 | 0.670 | 0.629 | 0.561 |
| RoBERTa-large-MNLI R | 0.275 | 0.347 | 0.237 | 0.166 | 0.795 | 0.743 | 0.690 | 0.625 |
| RoBERTa-large-MNLI F | 0.395 | 0.404 | 0.390 | 0.298 | 0.744 | 0.711 | 0.664 | 0.594 |
| DeBERTa-large P | 0.390 | 0.356 | 0.430 | 0.347 | 0.729 | 0.725 | 0.677 | 0.615 |
| DeBERTa-large R | 0.217 | 0.320 | 0.203 | 0.135 | 0.812 | 0.771 | 0.710 | 0.655 |
| DeBERTa-large F | 0.306 | 0.368 | 0.309 | 0.232 | 0.794 | 0.767 | 0.712 | 0.650 |
| DeBERTa-large-MNLI P | 0.470 | 0.335 | 0.369 | 0.397 | 0.721 | 0.710 | 0.673 | 0.608 |
| DeBERTa-large-MNLI R | 0.273 | 0.320 | 0.204 | 0.177 | 0.814 | 0.767 | 0.711 | 0.645 |
| DeBERTa-large-MNLI F | 0.369 | 0.354 | 0.282 | 0.278 | 0.796 | 0.760 | 0.714 | 0.643 |
| BART-large P | 0.333 | 0.365 | 0.412 | 0.270 | 0.776 | 0.769 | 0.708 | 0.643 |
| BART-large R | 0.189 | 0.308 | 0.168 | 0.092 | 0.825 | 0.788 | 0.700 | 0.639 |
| BART-large F | 0.256 | 0.357 | 0.268 | 0.164 | 0.824 | 0.796 | 0.718 | 0.653 |
| BART-large-MNLI P | 0.399 | 0.375 | 0.430 | 0.329 | 0.763 | 0.763 | 0.694 | 0.632 |
| BART-large-MNLI R | 0.213 | 0.317 | 0.169 | 0.111 | 0.823 | 0.785 | 0.694 | 0.633 |
| BART-large-MNLI F | 0.303 | 0.374 | 0.278 | 0.204 | 0.823 | 0.796 | 0.713 | 0.648 |
| | | | | | | | | |
| Best of Repurposed BERTScore | 0.470 | 0.411 | 0.483 | 0.397 | 0.825 | 0.796 | 0.719 | 0.660 |
| | | | | | | | | |
| **BERTScore, sentence-level, repurposed, using respective LMs below** | | | | | | | | |
| Cosine MPNet-base P | 0.436 | 0.218 | 0.226 | 0.365 | 0.721 | 0.755 | 0.665 | 0.642 |
| Cosine MPNet-base R | 0.224 | 0.257 | 0.086 | 0.121 | 0.745 | 0.740 | 0.608 | 0.553 |
| Cosine MPNet-base F | 0.375 | 0.279 | 0.164 | 0.272 | 0.764 | 0.768 | 0.647 | 0.598 |
| Cosine MPNet-base Sum-wt P | 0.435 | 0.214 | 0.233 | 0.370 | 0.736 | 0.770 | 0.671 | 0.639 |
| Cosine MPNet-base Sum-wt R | 0.339 | 0.282 | 0.134 | 0.248 | 0.730 | 0.720 | 0.631 | 0.608 |
| Cosine MPNet-base Sum-wt F | 0.411 | 0.265 | 0.193 | 0.327 | 0.778 | 0.787 | 0.678 | 0.636 |
| MNLI DeBERTa-large-MNLI E-C P | 0.526 | 0.182 | 0.099 | 0.406 | 0.457 | 0.487 | 0.489 | 0.485 |
| MNLI DeBERTa-large-MNLI E-C R | 0.233 | 0.163 | -0.078 | 0.161 | 0.437 | 0.381 | 0.418 | 0.393 |
| MNLI DeBERTa-large-MNLI E-C F | 0.269 | 0.184 | -0.046 | 0.190 | 0.280 | 0.211 | 0.263 | 0.200 |
| MNLI DeBERTa-large-MNLI E-C Entropy-wt P | 0.561 | 0.202 | 0.147 | 0.426 | 0.404 | 0.445 | 0.454 | 0.454 |
| MNLI DeBERTa-large-MNLI E-C Entropy-wt R | 0.004 | 0.011 | -0.020 | -0.038 | 0.230 | 0.225 | 0.264 | 0.264 |
| MNLI DeBERTa-large-MNLI E-C Entropy-wt F | -0.099 | -0.011 | -0.025 | -0.102 | 0.037 | -0.008 | 0.060 | 0.049 |
| | | | | | | | | |
| **Repurposed other metrics** | | | | | | | | |
| ROUGE-1 R | 0.162 | 0.157 | -0.011 | 0.054 | 0.779 | 0.709 | 0.621 | 0.558 |
| ROUGE-2 R | 0.298 | 0.189 | 0.045 | 0.190 | 0.788 | 0.719 | 0.643 | 0.590 |
| ROUGE-L R | 0.296 | 0.211 | 0.100 | 0.185 | 0.788 | 0.714 | 0.650 | 0.588 |
| BLEURT | 0.221 | 0.270 | 0.366 | 0.217 | 0.606 | 0.586 | 0.612 | 0.588 |
| MoverScore | 0.184 | 0.252 | 0.137 | 0.104 | 0.675 | 0.611 | 0.596 | 0.549 |
| | | | | | | | | |
| **Baselines, other reference-free metrics** | | | | | | | | |
| Blanc | 0.259 | 0.224 | 0.089 | 0.172 | 0.731 | 0.680 | 0.619 | 0.587 |
| SummaQA | 0.248 | 0.186 | 0.115 | 0.157 | 0.588 | 0.553 | 0.507 | 0.474 |
| SUPERT | 0.393 | 0.257 | 0.132 | 0.296 | 0.766 | 0.774 | 0.651 | 0.579 |
| SueNes | 0.244 | 0.243 | 0.168 | 0.231 | 0.787 | 0.785 | 0.695 | 0.667 |
| ChatGPT (Wang et al., 2023) | 0.512 | 0.473 | 0.456 | 0.443 | 0.645 | 0.587 | 0.487 | 0.524 |
| SDC* (Liu et al., 2022) | -0.082 | -0.093 | 0.070 | 0.028 | -0.712 | -0.631 | -0.553 | -0.504 |
| | | | | | | | | |
| **Reference-based approaches used in their originally designated way** | | | | | | | | |
| ROUGE-1 R | 0.214 | 0.270 | 0.129 | 0.148 | -0.050 | -0.013 | -0.081 | -0.084 |
| ROUGE-2 R | 0.212 | 0.221 | 0.131 | 0.123 | -0.091 | -0.059 | -0.112 | -0.105 |
| ROUGE-L R | 0.178 | 0.221 | 0.153 | 0.133 | -0.086 | -0.054 | -0.107 | -0.104 |
| MoverScore | 0.180 | 0.268 | 0.099 | 0.132 | -0.031 | 0.008 | -0.056 | -0.068 |
| BERTScore RoBERTa-base P | 0.003 | 0.171 | 0.269 | 0.068 | -0.099 | -0.045 | -0.087 | -0.080 |
| BERTScore RoBERTa-Base R | 0.171 | 0.362 | 0.269 | 0.128 | 0.323 | 0.322 | 0.252 | 0.216 |
| BERTScore RoBERTa-Base F | 0.089 | 0.289 | 0.298 | 0.103 | 0.088 | 0.119 | 0.063 | 0.045 |
| BERTScore RoBERTa-large P | 0.023 | 0.201 | 0.325 | 0.100 | -0.070 | -0.020 | -0.044 | -0.032 |
| BERTScore RoBERTa-large R | 0.192 | 0.375 | 0.292 | 0.147 | 0.282 | 0.280 | 0.242 | 0.214 |
| BERTScore RoBERTa-large F | 0.112 | 0.310 | 0.347 | 0.134 | 0.077 | 0.107 | 0.075 | 0.068 |
| BLEURT | 0.021 | 0.208 | 0.176 | 0.046 | 0.075 | 0.101 | 0.026 | 0.018 |
| METEOR (Banerjee and Lavie, 2005) | 0.218 | 0.283 | 0.106 | 0.135 | 0.056 | 0.069 | 0.003 | -0.010 |

