# OpenReview forum: "DocAsRef: An Empirical Study on Repurposing Reference-based Summary Quality Metrics as Reference-free Metrics"
_EMNLP/2023/Conference — EMNLP 2023 Findings_

### Official Review · Reviewer_Je5A · 2023-08-03

**Typos Grammar Style And Presentation Improvements:** None
**Soundness:** 3

**Excitement:**

3: Ambivalent: It has merits (e.g., it reports state-of-the-art results, the idea is nice), but there are key weaknesses (e.g., it describes incremental work), and it can significantly benefit from another round of revision. However, I won't object to accepting it if my co-reviewers champion it.

**Missing References:**

None

**Paper Topic And Main Contributions:**

This paper proposed the idea of re-purposing the reference-based metrics to be used as reference-free metrics for summarization task. The input document is used as the second input instead of the reference summary. By experimenting on different signle-document summarization datasets, they showed that re-purposing approach imporves the performance, and outperforms the non-GPT baselines, and has comparable results with GPT3.5 while having 0.5B parameters.

**Questions For The Authors:**

QA: Can you explain the generality of the approach (e.g. in multi document summarization)?
QB: Please explain your clues about low numbers for reference-based metrics (Table 4 & 5)?

**Reasons To Accept:**

1. The repurposing approach improves the performance of reference-based metrics (and also making them reference-free) on multiple datasets.
2. It outperforms non-GPT baselines including reference-free and reference-based metrics.
3. It has comparable results with GPT3.5-based baseline while having 0.5B parameters.


**Reasons To Reject:**

1. The metric is just tested on one domain and one type (single document), so its generalization is not clear.
2. Results for reference-based metrics in result tables (e.g. Table 4 & 5) are very low, any clue?
2. The writing is not well-organized

**Reproducibility:**

5: Could easily reproduce the results.

**Reviewer Confidence:**

4: Quite sure. I tried to check the important points carefully. It's unlikely, though conceivable, that I missed something that should affect my ratings.

---

> ### Author Rebuttal · Authors · 2023-08-29
>
> We are encouraged that the reviewer found our method’s strong performance compared with non-GPT baselines and GPT-3.5.
>
> **Reason 1 to reject on generality**: A) **Domain**: Unfortunately, all human evaluation datasets for summarization are from the news domain. We'd be happy to learn and evaluate on any non-news summarization  datasets with human evaluation. B) **multi-document case**: We added summary-level, Spearman's CC on the TAC 2010 dataset below. **BERTScore-based metrics still outperform baselines on the Pyramid score and the overall score by 9% and 7% respectively.**
>
> |  | **Pyramid** | **Linguistic** | **Overall** |
> |---|---|---|---|
> | **Our Max** | **_0.539_** | **0.422** | **_0.500_** |
> | **Baseline Max** | **0.492** | **_0.460_** | **0.470** |
> | bertscore-deberta-large-mnli_precision | 0.496 | 0.401 | 0.455 |
> | bertscore-deberta-large-mnli_recall | **0.526** | _0.405_ | **0.492** |
> | bertscore-deberta-large-mnli_f1 | **_0.539_** | **0.422** | **_0.500_** |
> | bertscore-bart-large-mnli_precision | 0.471 | 0.272 | 0.415 |
> | bertscore-bart-large-mnli_recall | 0.422 | 0.202 | 0.380 |
> | bertscore-bart-large-mnli_f1 | 0.481 | 0.245 | 0.426 |
> | bertscore-roberta-large-mnli_precision | 0.469 | 0.306 | 0.418 |
> | bertscore-roberta-large-mnli_recall | 0.481 | 0.340 | 0.450 |
> | bertscore-roberta-large-mnli_f1 | _0.509_ | 0.356 | 0.464 |
> | **Baselines below**  |  |  |  |
> | Blanc | 0.427 | 0.294 | 0.397 |
> | summaQA-F1 | 0.301 | 0.243 | 0.286 |
> | SummaQA-confidence | 0.290 | 0.152 | 0.262 |
> | Supert | 0.479 | 0.324 | 0.427 |
> | SueNes | 0.492 | **_0.460_** | _0.470_ |
>
> **Reason 2 to reject and Question, the poor performance of ref-based metrics**: First, the ref-based metrics are not our work. The numbers of SummEval were obtained using the SummEval toolkit itself while the numbers on Newsroom were copied from SueNes (Bao et al., NAACL 2022).
>
> As to why, there are two possible reasons. First, refs in the SummEval and Newsroom datasets are very short, e.g, as short as one sentence in Newsroom. A very short reference provides limited information for a ref-based metric to function. On TAC, the refs are much longer and thus ref-based metrics have higher scores than ref-free ones.

---

### Official Review · Reviewer_2AuP · 2023-08-04

**Typos Grammar Style And Presentation Improvements:** 1. In Section 1, the excessive use of…
**Soundness:** 3

**Excitement:**

3: Ambivalent: It has merits (e.g., it reports state-of-the-art results, the idea is nice), but there are key weaknesses (e.g., it describes incremental work), and it can significantly benefit from another round of revision. However, I won't object to accepting it if my co-reviewers champion it.

**Missing References:**

TRUE: Re-evaluating Factual Consistency Evaluation (Honovich et al., NAACL 2022)

Refer to line 125, "While the idea of using 124 the document as the reference is not new". This work focuses on evaluating consistency and all metrics used take src and sys as input, which should be included in the bibliography.

**Paper Topic And Main Contributions:**

This paper revisits general reference-based metrics for summary evaluation, proposing that using source documents as references can improve the performance of reference-based metrics.

**Questions For The Authors:**

Can you provide more details about the creation of idf in the usage of BERTScore? From my experience, using traditional BERTScore without applying the idf strategy, only replacing ref with source could not acquire such a high Pearson score. Is the improvement all come from the selection of idf_dict?

**Reasons To Accept:**

This paper makes a modification upon the input of reference-based metrics like BERTScore, transforming them into reference-free form, which is a simple way to improve their performance.

**Reasons To Reject:**

1. As mentioned in the paper, the practice of using sources as references has been employed in various other metrics and is not limited to the summarization task, making it a less novel idea.

2. The lack of crucial experiment details reduces the persuasiveness of the results presented in the paper.

**Reproducibility:**

3: Could reproduce the results with some difficulty. The settings of parameters are underspecified or subjectively determined; the training/evaluation data are not widely available.

**Reviewer Confidence:**

4: Quite sure. I tried to check the important points carefully. It's unlikely, though conceivable, that I missed something that should affect my ratings.

---

> ### Author Rebuttal · Authors · 2023-08-29
>
> Thank you for the constructive feedback.
>
> Our **contribution and novelty** is not using new inputs. Such a choice of inputs has been used in all baseline ref-free summary quality metrics including those based on ChatGPT. Our contribution is on using a zero-shot and extremely simple approach to achieve a ref-free metric. On top of the idea, we performed extensive experiments using different foundation models and various settings to study the robustness of this approach. Notably, we found that this approach can achieve ChatGPT-level performance with only 0.5B parameters, as compared to 175B of ChatGPT. Finally, we suggest explanations as to why such a simple approach can be effective.
>
> Regarding **the  lack of crucial experiment details**, please kindly let us know any additional info you expect and we will be happy to add. Details on the foundation models, versions of BERTScore, etc. are already given in Section 3 and Appendix A. We further provided the source code in Appendix C and included a link to the anonymized code repo in Section 1.
>
> Regarding **IDF**, first, in BERTScore's original paper (Zhang et al. ICLR 2020), IDF makes very marginal impact on all except one datasets/tasks. IDF often slightly decreases the performance. In our case, **IDF=False for all results in the manuscript**. We further reran all experiments on BERTScore-P with IDF=True. Due to space limits, we present the Spearman's CC using three base-size foundation models below, with and without IDF, respectively. The outcome is the same as in BERTScore's paper that **IDF has a very limited impact**.
>
> **IDF=True, newly got for rebuttal:**
> |  | summeval |  |  |  | newsroom |  |  |  |
> |---|---|---|---|---|---|---|---|---|
> |  | con | rel | coh | flu | inf | rel | coh | flu |
> | RoBERTa-base P |  0.295 |  0.284 | 0.381 | 0.228 | 0.627 | 0.579 | 0.589 | 0.536 |
> | BART-base P | 0.279 | 0.283 | 0.359 | 0.208 | 0.673 | 0.631 | 0.664 | 0.620 |
> | DeBERTa-base P | 0.262 | 0.252 | 0.316 | 0.206 | 0.614 | 0.556 | 0.613 | 0.544 |
>
> **IDF=False, reported in our manuscript:**
>
> |  | summeval |  |  |  | newsroom |  |  |  |
> |---|---|---|---|---|---|---|---|---|
> |  | con | rel | coh | flu | inf | rel | coh | flu |
> | RoBERTa-base P | 0.307 | 0.315 | 0.408 | 0.240 | 0.597 | 0.551 | 0.579 | 0.531 |
> | BART-base P | 0.291 | 0.322 | 0.390 | 0.233 | 0.675 | 0.650 | 0.661 | 0.610 |
> | DeBERTa-base P | 0.281 | 0.276 | 0.345 | 0.221 | 0.628 | 0.587 | 0.631 | 0.586 |
>
> Some additional note: We cannot perform IDF weighting on BERTScore-Recall and thus BERTScore-F1 because in the recall version, a token's logarithmic IDF is 0 because it appears in all documents as all documents are the same, i.e., the source document. This is a theoretical limitation per BERTScore authors in their GitHub repo `Tiiiger/bert_score` issue 8.

---

### Official Review · Reviewer_j2Me · 2023-08-05

**Soundness:** 3

**Excitement:**

3: Ambivalent: It has merits (e.g., it reports state-of-the-art results, the idea is nice), but there are key weaknesses (e.g., it describes incremental work), and it can significantly benefit from another round of revision. However, I won't object to accepting it if my co-reviewers champion it.

**Missing References:**

As far as I could tell, the presented references are enough.


**Paper Topic And Main Contributions:**

This paper aims to offer a reference free metric derived from a reference based one generating a small scale language model that is competitive over state of the art transformer-based alternatives. The proposed repurposing of metrics are compared against reference-free baselines indicating competitive results for the proposed technique.


**Questions For The Authors:**

When you say that the repurpose a ref-basic metric into a ref-free one you say you feed the document instead of the reference, but this seems to violated the need of an independent train and test, potentially leading to an over-fitting situation. How can you be sure this is not happening?

**Reasons To Accept:**

The work presents a relevant contribution, and it is fairly well written. The proposed method is sound and the results indicate a promising approach.


**Reasons To Reject:**

The work is clearly still in progress and some further analysis as an ablation study needs to be conducted, but those are natural limitations of an exploratory study in its preliminary stages.


**Reproducibility:**

3: Could reproduce the results with some difficulty. The settings of parameters are underspecified or subjectively determined; the training/evaluation data are not widely available.

**Reviewer Confidence:**

3: Pretty sure, but there's a chance I missed something. Although I have a good feel for this area in general, I did not carefully check the paper's details, e.g., the math, experimental design, or novelty.

**Typos Grammar Style And Presentation Improvements:**

As far as I could tell, there were no typos or grammar issues.

---

> ### Author Rebuttal · Authors · 2023-08-29
>
> We appreciate the reviewer for finding our approach to be competitive over the SOTA alternatives.
>
> For the question for the authors, we would like to **clarify that our method is zero-shot** as mentioned in Pages 3 & 6. There is no training and thus no overfitting risk. The relationship between document and reference is not one for training and the other for test.
>
> We appreciate the feedback on **ablation study**. Kindly let us know any additional experiments you feel necessary and we will add them. The paper already included extensive results using different foundation models and settings to study the robustness of our approach and its independence to foundation models. The conclusion is that we can get consistent results.
>
> An additional ablation study about IDF was done in response to Reviewer 2AuP's concern. Please feel free to read the results there.

---

### Meta-Review · Area_Chair_cq7K · 2023-09-07

**Recommendation:** 3

**Metareview:**

This paper focuses on summarization evaluation and investigates whether reference-based metrics can be repurposed as reference-free metrics by substituting the reference with the input document. Overall, reviewers agree that this paper has merits (simple/sound approach, promising results) but also point out that 1/ it has a number of issues regarding clarity and that 2/ the experimental results are quite preliminary.  For 1/ I feel that the authors's responses provide some feedback to be included in a revised version of the paper. For 2/ I believe that this is not such an issue for a short paper as this format is adequate for describing a work in progress.

---

### Decision · Program_Chairs · 2023-10-07

**Decision:**

Accept-Findings

**Comment:**

This paper focuses on summarization evaluation and investigates whether reference-based metrics can be repurposed as reference-free metrics by substituting the reference with the input document. Overall, reviewers agree that this paper has merits (simple/sound approach, promising results) but also point out that 1/ it has a number of issues regarding clarity and that 2/ the experimental results are quite preliminary.  For 1/ I feel that the authors's responses provide some feedback to be included in a revised version of the paper. For 2/ I believe that this is not such an issue for a short paper as this format is adequate for describing a work in progress.